# Anxiety in Nursing Students during the COVID-19 Pandemic: Systematic Review and Meta-Analysis

**DOI:** 10.3390/healthcare12161575

**Published:** 2024-08-08

**Authors:** Aroa García-Rivas, María Begoña Martos-Cabrera, María José Membrive Jiménez, Raimundo Aguayo-Estremera, Nora Suleiman Martos, Luis Albendín-García, José L. Gómez-Urquiza

**Affiliations:** 1Hospital Quirón Salud Sagrado Corazón, C. Rafael Salgado, 3, 41013 Sevilla, Spain; 2San Cecilio Clinical University Hospital, Av. Del Conocimiento s/n, Andalusian Health Service, 18071 Granada, Spain; 3Faculty of Health Sciences, University of Granada, 18017 Granada, Spain; 4Department of Psychobiology and Methodology in Behavioral Sciences, Complutense University of Madrid, Campus de Somosaguas, 28223 Pozuelo de Alarcón, Spain; 5Casería de Montijo Health Center, Granada Metropolitan District, Andalusian Health Service, Calle Joaquina Eguaras nº 2, Edificio 2 1ª Planta, 18013 Granada, Spain; 6Instituto de Investigación Biosanitaria (ibs.GRANADA), 18012 Granada, Spain

**Keywords:** anxiety, COVID-19, nursing students, pandemic, risk factor prevalence

## Abstract

Objectives: To analyse the influence of the COVID-19 pandemic on the development of anxiety in nursing students and the factors involved. Design: A systematic review and meta-analysis. Data source: PubMed, CINAHL, Scopus and Web of Science. Background: Nursing students are at an increased risk of developing mental overload, due to the presence of many sources of stress during their academic training. Therefore, the COVID-19 pandemic has had an impact on the mental health of the general population, especially on healthcare workers and consequently on students undertaking placements in healthcare settings. Methods: A systematic review was conducted using PubMed, CINAHL, Scopus and Web of Science databases. A total of 24 articles were included in the review, and 20 articles were selected for the meta-analysis. Results: We found that the anxiety scores of nursing students during the COVID-19 pandemic were slightly higher (50%) than before the pandemic. The most influential risk factors for developing anxiety were academics, age, gender, having children, living in urban areas or with family, having an addiction to social networks, and having a fear of becoming infected with COVID-19. Resilience, spiritual support and feelings of happiness protected students against the risk of developing high levels of anxiety. Conclusions: The COVID-19 pandemic has led to increased levels of anxiety in nursing students. Thirty-five percent of the meta-analytically analysed sample had elevated levels of anxiety.

## 1. Introduction

According to the World Health Organisation (WHO), mental health is defined as “a state of well-being in which a person realises his or her capabilities and is able to cope with the normal stresses of life, to work productively and to contribute to his or her community” and is influenced by psychological, biological and social factors [1]. The International Statistical Classification of Diseases and Related Health Problems (ICD-11) describes mental and behavioural disorders, classifying them into different disorders according to problems with thought processes, emotions, behaviour and relationships with others [1]. In order to diagnose a mental disorder, signs and symptoms must appear repeatedly and affect one or more spheres of daily life [2]. In this way, an alteration in behaviour or affectivity in isolation does not determine a mental illness. One of the symptoms most commonly linked to the term mental health is stress. In recent years, this term has gained great interest among researchers, discovering that it is one of the most prevalent factors in our community [3]. When we talk about stress, we refer to the biological alert system in the face of imminent danger, which allows us to survive [4]. Throughout the day, we are confronted with several stimuli that activate the sympathetic nervous system, which increases stress levels in the organism. After the cessation of the dangerous stimulus, our organism returns to basal stress levels. The problem arises when the stimulus, which our body interprets as harmful, remains continuously over time or when the individual is not able to handle the stress. This would provoke a more advanced physical and psychological effect, which would lead to functional and organic problems related to emotional fatigue [5]. Stress sustained over time can condition the person who suffers from it, leading to major mental health disorders such as anxiety, insomnia or depression. Although stress is closely related to these pathologies, it is not the only etiological factor [3].

In the scientific literature, anxiety is the most predominant variable in the study of mental health [6]. This variable is defined as excessive restlessness in different situations of everyday life [7]. On a daily basis, we live with baseline levels of anxiety, which allow us to be alert to environmental disturbances [8]. But if these levels are chronically elevated, that condition can have a detrimental impact on the individual’s quality of life. In this case, it would be diagnosed as an anxiety disorder [9]. If anxiety manifests itself over a long period of time, it will refer us to symptoms similar to those of sustained stress, both mentally and physically [10].

Among the professions most affected by stress are those of health professionals. Among these professionals are nurses, whose training has evolved over the years. In the course of time, nursing achieved a holistic approach, which goes beyond the care of the individual and is directed towards the person, the environment and health [11]. Nurses are the health professionals who spend the most time caring for the sick and providing them with health education [12]. For this reason, these healthcare professionals are considered one of the groups at greatest risk of suffering from mental health disorders, highlighting problems of anxiety and even the development of burnout [13,14,15]. The main causes of this phenomenon among nurses are the shortage of qualified professionals, work overload [16], the numerous competencies they have or the pressure to provide care throughout their working day [17,18]. All this leads to an increased risk of developing Burnout Syndrome, job dissatisfaction and the desire to change jobs [11,19].

In general, university students are subjected to numerous sources of stress in the academic environment. However, the stress faced by nursing students exceeds these levels, since in addition to theoretical learning, they must attend clinical practice in the care environment, which is an additional stressor for students [20]. Nursing curricula require a high level of knowledge and skills acquisition in the care setting [21]. In addition, students must provide quality and respectful clinical care to patients [19] and continuously deal with patients’ suffering, pain or death [22]. During clinical placements, each student is mentored by a nursing professional, who must pass on his or her knowledge and train students in care work [23]. This period is crucial, since, as described by Lepiani-Díaz et al. [24] and Devi et al. [25], the beginning of clinical practice in nursing is the main cause of moderate to high levels of stress and anxiety, which negatively affect both the physical and psychological health of students and are also related to low levels of clinical performance.

If we combine the stress and anxiety developed by the need to combine university life with care practices and the current situation of nurses in clinical settings [26], we will find students with an increased risk of developing anxiety and burnout [27,28]. If these alterations in mental health are maintained over time, they could affect motivation, learning and the acquisition of clinical skills, causing students to provide lower quality care to patients in the service where they are rotating [29].

Coronavirus disease 2019 (COVID-19), which appeared in December of that year in Wuhan (China), was responsible for the most important respiratory infection in recent years and was upgraded to pandemic status by the WHO in March 2020 [30,31]. Throughout history, pandemics have significantly affected the health of millions of people, in addition to causing major global crises [32]. Consequently, an increase in mental disorders in the population is predicted [33].

With the advent of the COVID-19 pandemic, nurses were overwhelmed in terms of workload, as they were the mainstay of care for patients with the infection [34,35]. This led to an increase in mental disorders and consequently burnout [36].

Because of the pandemic and health overload, nursing students underwent a drastic change in their training, decreasing interaction and communication with other students and university professors, affecting their academic performance [37,38]. These changes increased the workload and affected the knowledge and skills they had to acquire to perform their clinical work during clinical placements. In addition, some students suffered COVID-19 infection during their hospital internships, when confinement had not yet been declared. They also had to care for seriously ill patients, who died in the hospital without any possibility of being cured, since the evolution or treatment of this infection was not well known [39,40]. This fact promoted a greater emotional impact, generating an increased risk of suffering high levels of anxiety and other mental disorders among nursing students [41].

Once the impact of the pandemic on nursing professionals and students of this degree had been observed, it was proposed to analyse the prevalence of anxiety among those students during the COVID-19 pandemic, as well as the primary factors contributing to increased anxiety during the pandemic and protective factors that help to mitigate anxiety in nursing students.

## 2. Materials and Methods

### 2.1. Study Design and Search Strategy

A systematic review with meta-analysis was conducted, following the standards set by the PRISMA statement [42].

Firstly, some keywords were chosen in the Medical Subject Headings (MeSH) thesaurus, the Boolean operators “OR” and “AND” were added, and finally, the search strategy to be used in the exploration of articles was obtained: “(undergraduate nurses OR nursing students) AND anxiety AND (COVID-19 OR pandemic COVID-1 OR SARS-CoV-2)”. This search strategy was used in the following databases: PubMed, WOS, Scopus and CINAHL. The literature search was conducted in November 2023 and February 2024.

### 2.2. Inclusion and Exclusion Criteria

In order to start the search for articles, a series of inclusion criteria were established to help us choose the right articles for our bibliographic review. The inclusion criteria chosen were (a) publications less than or equal to 5 years old; (b) no restrictions on the language available; (c) quantitative studies; (d) publications measuring the variable anxiety in nursing students; and (e) validated measurement instruments.

### 2.3. Selection of Studies

First, three members of the research group carried out a literature search by applying the selected filters in the various databases. The resulting number of articles for title and abstract reading was 520 articles (Figure 1). Duplicate articles and those that did not meet the inclusion criteria were eliminated. Subsequently, these same researchers critically read the studies independently. Thus, if there were discrepancies between these researchers, a fourth member of the group intervened to reach a consensus on the selection of articles and to detect the presence of possible methodological errors.

### 2.4. Data Analysis and Coding

The variables analysed in the selected publications were as follows: (a) bibliographic data (authors, year and country of publication); (b) study design; (c) sample; (d) measurement instruments; (e) time of measurement; (f) mean, standard deviation, prevalence or median of the anxiety variable; (g) significant results related to the study variables; and (h) degree of recommendation and levels of scientific evidence according to the Oxford Centre for Evidence-Based Medicine (OCEBM). These variables have been organised in Table 1.

### 2.5. Assessing Study Quality and Determining Bias

The methodological quality of each study was analyzed by different members of the group; in case of disagreement, another member of the group intervened. The statistical software Stats Direct (Version 4) was used to perform the meta-analysis. First, a sensitivity analysis was performed. Subsequently, the absence of publication bias was assessed using Egger’s linear regression. The I2 index was used as a measure of heterogeneity (99.3%).

## 3. Results

Following the inclusion criteria, 62 articles were selected for abstract reading, then a total of 42 articles were selected for full-text reading. Finally, a total of 28 articles were included in the systematic review, and 20 in the meta-analysis.

### 3.1. Characteristics of the Included Studies

Most of the articles included in the review were conducted in Turkey [34,43,45,50,52,53,57,59,64], Israel [40,54,62], the United States [38,46] or Spain [48,51].

The chosen studies had a descriptive cross-sectional design, although some prospective longitudinal studies [48,51,54] and a randomized clinical trial [57] were also included.

As for the sample, the majority of the students were female. The most commonly used measurement instruments were the Generalized Anxiety Scale (GAD-7) [43,46,49,54,58,60], Beck Anxiety Inventory (BAI) [34,50], Trait Anxiety Inventory (STAI) [50,52,59,61] and the Coronavirus Anxiety Scale (CAS) [55,64].

Most of the studies were evaluated in 2020 because the COVID-19 pandemic was established in the same year. However, two of them studied the population also in years before the pandemic [46,48].

### 3.2. Pre-Pandemic COVID-19 Anxiety Levels in Nursing Students

According to the first measurement, conducted in 2017 by authors Reverté-Villarroya et al. [48] in Spain, 35.9% of nursing students scored above the threshold defined for elevated anxiety levels.

Similarly, Kim et al. [46] conducted a measurement just before the confinement caused by the COVID-19 pandemic. These authors analysed how 19.4% of the students surveyed scored greater than or equal to 10 on the GAD-7 scale, a threshold that measures elevated levels of anxiety.

Both results show a low prevalence of anxiety among nursing students before the COVID-19 pandemic.

### 3.3. Levels of Anxiety in Nursing Students during the COVID-19 Pandemic

According to the results obtained by Reverté-Villarroya et al. [48], the COVID-19 pandemic increased the risk of developing anxiety in nursing students. This could be confirmed by the results obtained from the analysis of the selected studies, which showed prevalences ranging from 51.9% [50,52] and 55.2% [46], to 59.7% [48] of nursing students who scored high for the development of anxiety during the COVID-19 pandemic. However, in a study conducted in the United States, the prevalence of anxiety due to the pandemic was as high as 70% among students, according to the study conducted by Fitzgerald and Konrad [38]. Similarly, anxiety levels in the students surveyed rose during the months of the pandemic, being higher during the last weeks of confinement [51].

If we stratify by anxiety levels, we find that the majority of nursing students surveyed showed mild non-pathological anxiety towards COVID-19, specifically 32.2% [43], 53.89% [49], 64.09% [59] and 66.9% [45].

According to a study conducted in Israel, 42.8% of students scored at moderate levels of anxiety during the COVID-19 pandemic. This is consistent with the results of research in Turkey, with scores of 39.2% [34] and 40.5% [43], but far higher than the results for moderate anxiety obtained by other researchers, of 16.5% [45], 21.3% [59] and 27.1% [49].

On the other hand, for severe levels of anxiety, the results were uneven across the different studies reviewed, with prevalences ranging from 13.1% [51], 13.8% [60], 16% [62], 16.6% [45], 19% [49], 22.5% [43], 35.3% [63], up to 51.5% [34].

In contrast to the above results, Limugha et al. [47] in India measured anxiety during the last months of confinement. They concluded that 96% of respondents reported functional anxiety in the face of the COVID-19 pandemic, while only 4% showed dysfunctional or pathological levels of anxiety. The latter is consistent with the results obtained by Mundakir et al. [55], in which only 6% of the sample showed elevated levels of anxiety, as in the study published by Sun et al. [44], in which they stated that the percentage of students with total anxiety was 12.46%.

### 3.4. Risk Factors That Influenced the Development of Anxiety during the COVID-19 Pandemic in Nursing Students

The main factors that negatively impacted the mental health of nursing students during the COVID-19 pandemic were associated with highly stressful situations, related to academic activities [38], subject assessment [48], the suspension of clinical practice [49] or the transition from face-to-face classes to online transmission [34,38,49]. After months of confinement, when returning to face-to-face classes at the university, most of the nursing students showed an increase in anxiety levels related to the risk of COVID-19 infection [61,62,64].

In addition, different authors analysed the influence of the academic year in which the nursing students were studying during confinement on the development of anxiety. Final-year students were the most affected [48,51], as were students from private universities [55]. They concluded that upperclassmen were twice as likely to develop mental health problems and the rise in anxiety levels was more pronounced throughout the confinement and faster than in the rest of their peers [48,51]. This does not support the results provided by Kuru Alici and Ozturk Copur [34] and Sun et al. [44], who concluded that first-year students showed higher levels of anxiety, while Limugha et al. [47] and Savitsky et al. [40] found no significant differences between the two variables in their research.

If we focus on the nursing profession, those students who had a negative view of the nursing profession during the pandemic and those students who saw themselves as incapable of practising nursing in the future [50] showed a greater predisposition to develop anxiety. However, if we compare the prevalence of anxiety between graduate nurses and nursing students, we find that nurses who were practising nursing during the COVID-19 pandemic showed higher percentages [61].

Students who were both working and studying and those who suffered from a lack of protective equipment or did not take sufficient infection prevention measures showed higher levels of anxiety [40,44,61,62,63,64,65].

In terms of demographic variables, men showed a lower likelihood of developing anxiety towards COVID-19 than women [34,40,45,51,54,59,61,63,65]. However, Sun et al. [44] found that male students surveyed showed a greater predisposition to develop anxiety, and the rise in anxiety levels during confinement was more pronounced in these students [51]. Anxiety levels were evened out between both genders in the last weeks of confinement [54].

Another risk factor related to the development of anxiety in nursing students during the pandemic was age. Younger students had higher scores on measures of anxiety levels [48,63,66].

Also, students who had dependent children or felt overwhelmed by the suspension of school and high school classes during the pandemic [40,54] showed a higher risk of developing pathological anxiety related to COVID-19.

Focusing on the sociodemographic factors of nursing students, a significant relationship was found between the development of anxiety and the presence of family problems, financial problems, social distancing and restrictions on social life [43,62]. Those students who lived with their family members during confinement showed higher levels of anxiety [45]. Furthermore, these levels increased exponentially as a function of the number of people living in the family home [45]. According to Kuru Alici and Ozturk Copur [34], students who lived in urban areas during the pandemic were more likely to suffer from anxiety than those who lived in rural areas or had a garden at home [51]. Similarly, students who had to be in isolation or quarantine due to contact with a COVID-19 patient, those who had symptoms compatible with the infection [53], or who had family members with such symptoms were more likely to develop anxiety [34]. In contrast, Limugha et al. [47] and Sakai et al. [56] failed to significantly associate sociodemographic variables with elevated anxiety levels.

In terms of the most influential psychological variables, we found that students with high scores on the different scales measuring anxiety were those with a greater fear of COVID-19 [34,40,49,50,52,58,61,62,64,65]. Similarly, students with higher levels of anxiety during the pandemic showed the following associated feelings: sadness, worry, rage, anger, irritability, problems relaxing and depressive symptoms, nervousness, excessive uncontrollable worry, restlessness and fear [49,58,59].

Also, a statistical relationship was found between sleep quality and anxiety status of nursing students [52]. Thus, students who slept fewer hours, ate less or had dysfunctional coping strategies were more likely to develop anxiety during the pandemic [52,53,60,67].

Regarding the spirituality and religion variables, a study in Israel [54] showed that Muslim students were at higher risk of developing anxiety at the beginning of the pandemic, with increasing levels throughout the confinement, whereas Jewish and Christian students showed a lower risk of developing anxiety from the beginning of the pandemic, and their levels decreased throughout the confinement [54]. However, the authors Limugha et al. [47] in India failed to find significant relationships between the spirituality/religion variables and elevated levels of anxiety.

On the other hand, students who consulted information on social networks about the pandemic and the symptoms they felt, were exposed to over-information, or manifested a possible internet addiction showed a higher predisposition to develop COVID-19 anxiety during confinement [54,62,64] and to suffer from cyberchondria [54]. The latter, in turn, is directly related to increased anxiety levels [54]. Elevated anxiety caused by the pandemic acted as a mediating variable in the relationship between internet addiction and the development of cyberchondria [54].

### 3.5. Protective Factors against the Development of Anxiety by Nursing Students during the COVID-19 Pandemic

Psychological factors were most influential in protecting against increased levels of anxiety. Resilience, feeling happy and good family functioning decreased anxiety levels. Spiritual support also acted as a mediator in managing anxiety in the face of the COVID-19 pandemic [46,54,59].

On the other hand, a clinical trial was conducted in Turkey, whose interventions were based on online sessions of laughter therapy, gamification, deep breathing and physical exercise during the COVID-19 pandemic [57]. These interventions improved the depressive symptoms experienced by the nursing students but failed to improve anxiety levels and feelings of loneliness experienced during confinement [57,65,66].

### 3.6. Meta-Analysis

According to Egger’s test, no publication bias was found in any of the cases. A total of 7742 nursing students were included in the meta-analysis, of whom 3318 reported anxiety due to the COVID-19 pandemic. The meta-analytic estimation of the percentage of nursing students with high or severe levels of anxiety was 35% (95% CI 23–49%) (Figure 2).

## 4. Discussion

First, we analysed the results obtained in longitudinal studies, which provided data measured before the onset of the COVID-19 pandemic. We found that the percentage of nursing students suffering from high levels of anxiety ranged from 19.4% [46] to 35.9% [48]. Data showing a slightly lower percentage than the figures reported by other articles [66,67] allowed the conclusion that a high percentage of nursing students suffered from stress [68,69,70,71] and severe anxiety during some trimester of the degree [72,73,74,75], especially coinciding with the start of clinical practice during the last trimesters of the degree [12,19,48,73,74,76].

Subsequently, the percentage of nursing students experiencing anxiety in the aftermath of the COVID-19 pandemic was examined. This was slightly higher than previous measurements, with percentages ranging around 50 [46,48,50,52]. If we stratify according to anxiety levels, we find that the meta-analytic study conducted for a sample of 7742 nursing students showed that the percentage of students with high or severe levels of anxiety was 35%. This is consistent with the studies included in the systematic review, whose results ranged from 13.1% [40] to 51.5% [34]. It can therefore be deduced that the COVID-19 pandemic has been a major source of anxiety for university students and to a greater extent for medical and health science students [75,77,78]. This increase could be mainly due to the change in the routine and lifestyle of students prior to confinement, although the increase in the prevalence of anxiety, depression and post-traumatic stress [76,79,80] has been more pronounced in graduate nurses who worked in the hospitals at the beginning of the confinement. This is demonstrated by the study conducted by Kang et al. [80,81], in Wuhan, where they concluded that 28% of nurses working on the frontline had moderate to severe levels of anxiety. The increase in the percentage of students suffering from anxiety, depression or stress [58] could have been associated with the feeling of hopelessness and sleep disturbances [82].

The risk factors that affected the development of pathological anxiety levels in nursing students during the COVID-19 pandemic were stratified into sociodemographic, psychological, spiritual and work/academic factors, the latter being the most influential [34,38,48,49,50]. The students who were most at risk of developing high levels of anxiety were those who had had their care practice suspended, and were in the final years of their degree [48,51], belonged to private universities, had not been able to adapt to online classes [55] and those who had witnessed the lack of personal protective equipment during the internship [40,44,60,66]. Students in their final years of clinical practice may have been more vulnerable to developing anxiety and thus less motivated to complete their studies and work as nurses, due to the high exposure to the virus in this profession, when their clinical practice was suspended. Regarding the change in teaching modality, studies conducted in different developed countries concluded that health students were more satisfied with online classes, as it allowed them to save time [80,81,82,83,84,85].These differences in online classes could be explained by a study conducted in Spain, which concluded that adult students with a family burden or difficult access to new technologies have a higher risk of developing high levels of anxiety [86].

Within the sociodemographic variables, we found that women, in general, presented a higher risk of developing elevated levels of anxiety during the pandemic [34,40,45,51,54,59,60,63,64,67]. This result could be because females have a higher vulnerability to develop anxiety and fear [82,85], especially regarding COVID-19, as discussed by some authors [83]. Students of younger ages [48,63,67] and those students who had dependent children showed a higher risk of developing pathological anxiety toward COVID-19 [40,54]. These data confirm studies in China by Xiong et al. [33] and Ahmed et al. [87,88,89], in which they concluded that students aged 12–30 years were at higher risk of developing anxiety, stress and depression.

Students who lived with their family members during confinement [43,45,62,64], lived in urban areas [34], had symptoms compatible with the infection [53] or had family members who were quarantined due to contact with a COVID-19 patient or who had family members with such symptoms, were more likely to develop pathological anxiety [34]. However, a study conducted in China [90] concluded that university students who had experienced confinement with their parents had fewer anxiety symptoms. On the other hand, Brooks et al. [91] confirmed how the process of quarantine due to contact with a COVID-19 patient was positively related to the development of mental disorders, due to the loss of freedom, concern about the possible development of symptoms and, above all, due to the separation from close family members.

The most influential psychological factors were fear of COVID-19 [34,40,49,50,52,60,62] and the presence of dysfunctional coping strategies [52,53,60]. The most frequent symptoms associated with anxiety in students were irritability and excessive worry [49,58].

The most over-informed students, who tended to seek information about in the pandemic on social networks or had some internet addiction [54,62,64], were more likely to have elevated levels of anxiety. This is consistent with the results obtained by Sauer et al. [92], who found that students with high levels of hypochondriasis had higher levels of anxiety toward SARS-CoV-2. Currently, social networks serve as a source of information for students [93], especially since the confinement, during which younger students began to show greater attention to global news [94].

In terms of protective factors against increased anxiety levels in nursing students during the COVID-19 pandemic, we found that resilience, feelings of happiness, spiritual support and good family functioning acted as a barrier to the risk of pathological anxiety levels [46,54,59]. The presence of these factors acts to support students’ mental stability in the presence of unexpected new challenges [90,95,96]. Resilience is one of the intrinsic coping strategies that students develop in the face of stressful situations, such as the onset of confinement due to the COVID-19 pandemic and the consequent suspension of face-to-face classes [97]. Moreover, religiosity and spirituality may have acted by minimizing the consequences of social isolation in students. Some articles have shown that lower levels of worrying, sadness, depression, stress and fear of COVID-19 infection were associated with greater private religious activities, religious attendance, spiritual growth and with an increase in religious activities [98,99].

The implications of this study for the training of future nurses stem from the need to work on and improve the coping mechanisms for dealing with the stressful situations these students will have to face. The COVID-19 pandemic and the subsequent confinement have been one of the biggest disruptions to the stability of student nurses. However, these students still must cope with clinical placements, where the risk of exposure to the SARS-CoV-2 virus is still present. Furthermore, the proper functioning of these placements continues to suffer from the consequences of the pandemic, as they could be suspended again if levels of infection in the population increase again. On the other hand, it is necessary to reinforce the training of nursing students in the use of personal protective equipment, the different types of hospital isolation, and measures to prevent the spread of infections in both healthcare and university settings. It is also essential to include practical sessions in the training of students on how to act in emergency situations, as well as training them in protocols and measures to protect their mental health in situations such as these.

The limitations of this study were, firstly, the small number of scientific articles published, as most of the publications analysed health professionals or students in general, without being focused specifically on nursing students. Secondly, most of the studies found were observational studies, preventing us from establishing causal relationships between the different variables and the COVID-19 pandemic. Due to the limited sample size, it is recommended to adopt these results with caution. Furthermore, nursing education is very heterogeneous from one country to another, being considered in some countries as a university degree, thus introducing greater variability in the results.

## 5. Conclusions

The COVID-19 pandemic and subsequent confinement led to an increase in the prevalence of anxiety disorders among student nurses. Thirty-five percent of the students selected for meta-analyses had elevated levels of anxiety.

The risk factors that most influenced the development of anxiety during the pandemic were stress caused by online academic activities and high exposure to pandemic information on social media; suspension from nursing practice; uncertainty about the future of the profession; and confinement. Sociodemographic factors that mediated the development of higher levels of anxiety in nursing students were age, gender, having children, presence of family support, and living at home or in urban areas. Students who were afraid of becoming infected or had symptoms of the infection showed a higher tendency to develop anxiety.

Resilience, feelings of happiness, spiritual support and professed faith in the Christian or Jewish religion showed extra protection against increased anxiety levels during confinement.

To create a risk profile in nursing students who are more susceptible to developing mental problems, related to stressful situations such as the COVID-19 pandemic, and to adapt teaching and educational environments to the characteristics of the students, this paper would help to determine the protective factors against stressors such as the COVID-19 pandemic in nursing students and the risk factors generating anxiety in undergraduate nursing students.

For future research, it is recommended to carry out a psychological analysis of students who were surveyed during the years of the pandemic and compare the levels of anxiety they currently present with those described in the research and develop additional knowledge on the psychological impact that the COVID-19 pandemic had in the long term on nursing students.

## Figures and Tables

**Figure 1 healthcare-12-01575-f001:**
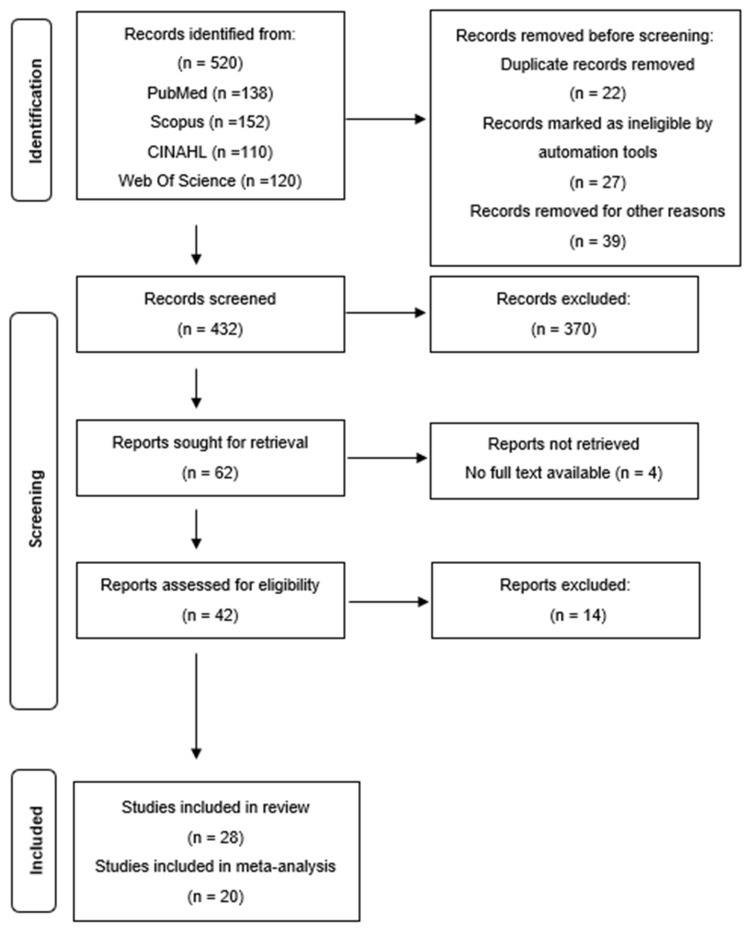
Study selection.

**Figure 2 healthcare-12-01575-f002:**
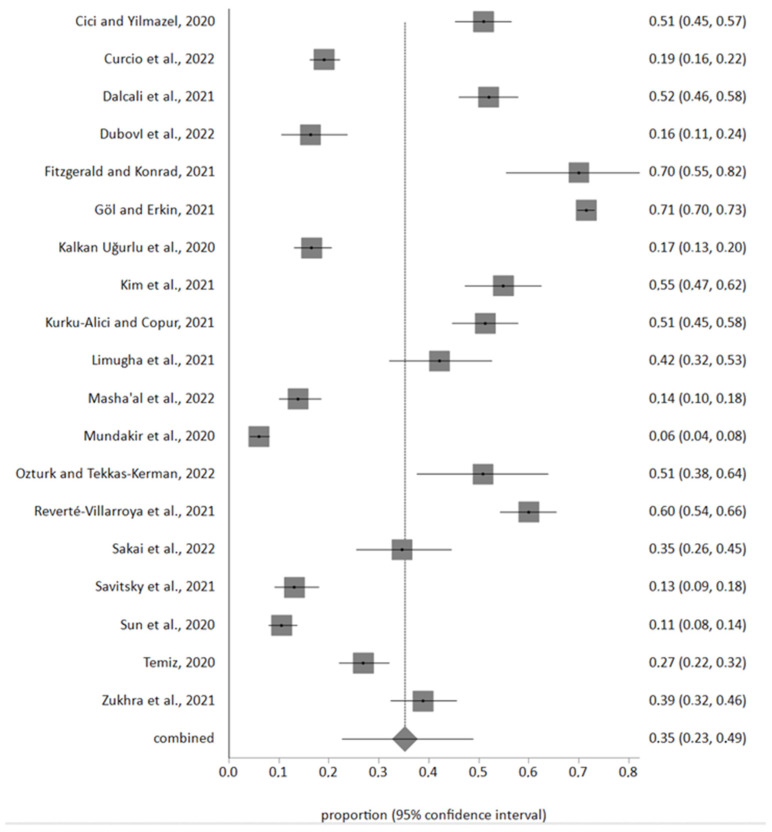
Forest plot of prevalence of severe levels of anxiety in nursing students [34,38,43,44,45,46,47,48,49,50,52,53,54,55,56,57,62,63,67].

**Table 1 healthcare-12-01575-t001:** Main results.

Author, Year and Country of Publication	Design/Sample	Measuring Instrument	Anxiety Level Time of Measurement		Significant Results	LE/GR
			Before	During Pandemic		
Savitsky et al. [40], Israel, 2020	Cross-sectional study n = 244 (Women 89.2%)	GAD-7		Moderate 42.8%; Severe 13.1%	Males had significantly lower anxiety scores (median = 7.0, IQR: 1.0–11.0) in comparison with females (median = 9.0, IQR: 5.25–14.0). The anxiety score increased among those students who were the parents of children, with the increase in the burden following the lack of educational frameworks for children in schools or kindergartens. The anxiety score of students who reported intense fear of infection was found to be significantly higher. Academic year of study was not found to be associated with level of anxiety. Lack of protection equipment among working students was found to be associated significantly with a higher anxiety score.	2c/B
Temiz [43], Turkey 2020	Cross-sectional study n = 316 (Women 72.2%)	GAD-7		Mild 32.3%; moderate 40.5%; high 22.5%; severe 4.7%.	High levels of anxiety were related to family, educational, economic, health and health problems and to the restriction of social life during COVID-19.	2c/B
Sun et al. [44], China, 2020	Cross-sectional study n = 474 (Women 84.8%)	SAS		Mean score ± SD: 39.54 ± 8.70 Prevalence of total anxiety: 12.4%.	Male students were more likely to suffer from anxiety disorders than female students (OR: 2.39; 95% CI: 1.26~4.52; *p* = 0.008). Nursing students in their sophomore year were more likely to suffer from anxiety disorders (OR: 5.30; 95% CI: 1.61~7.45; *p* = 0.006). Students with low levels of prevention behavior were more likely to experience anxiety (OR: 3.49; 95% CI: 1.16~5.19; *p* < 0.001).	2c/B
Kalkan-Uğurlu et al. [45], Turkey, 2020	Cross-sectional study n = 411 (Women 79.3%)	DASS-42		Normal 49.1%; mild 17.8%; moderate 16.5%; severe 7.1%; very severe 9.5%.	Female students had higher stress levels. The students who stayed with their family during the social isolation process had higher stress levels. As the number of people in the household increased, the anxiety level increased.	2c/B
Kim et al. [46], USA, 2021	Cross-sectional study n = 173 (Women 93.1%)	GAD-7	Level moderate to severe anxiety: 19.4%	Level moderate to severe anxiety: 55.2%	Anxiety levels increased during confinement. Resilience and good family functioning improved anxiety levels. Spiritual support acted as a mediator in the development of anxiety.	2c/B
Kuru Alici and Ozturk Copur, [34], Turkey, 2021	Cross-sectional study n = 234 (Women 67.9%)	BAI		3% very low 39.2% moderate 51.5% severe BAI score 26.56 ± 8.86	The BAI score was found to be higher in female students than in males. First-year students might suffer more severe anxiety than the other students. Students living in urban areas scored higher for anxiety. The anxiety level of nursing students was positively correlated with the fear of COVID-19. Students who had experienced home quarantine during the pandemic had more anxiety. Students whose family or relatives had COVID-19 had higher levels of anxiety. Students who were not satisfied with the remote learning had more anxiety.	2c/B
Limugha et al. [47], India, 2021	Cross-sectional study n = 95 (Women 92.6%)	CAS		96% functional anxiety 2% possible dysfunctional anxiety 2% dysfunctional anxiety	The stress and anxiety variables could not be significantly related to the demographic variables: age, grade, gender and religion.	2c/B
Reverté-Villarroya et al. [48], Spain, 2021	Cohort study n = 305 (Women 86.5%)	GHQ-28	35.9% scored more than 23 on the GHQ-28 test. Median 19 CI [14,15,16,17,18,19,20,21,22,23,24,25,26,27,28]	59.7% scored more than 23 on the GHQ-28 test. Median 23 IQ [18–36.5]	During the pandemic, there was a greater percentage of students with high anxiety. The COVID-19 pandemic was statistically significant and increased the probability of having a higher total GHQ-28 score. Nursing students in the final year of their studies reported higher scores. The median age was observed to be a little lower in the group of participants with higher total scores (*p* = 0.046).	2b/B
Curcio et al. [49], Italy, 2022	Cross-sectional study N = 709	GAD-7		18.8% minimal; 35.09% mild; 27.1% moderate; 19% severe. Mean score: 9.46 ± 5.41	All the students reported scores referring to moderate levels of anxiety. The most common feelings experienced by students at high levels were sadness, worry, anger, irritability, fear of contagion, problems with relaxation, and depressed mood. Some students indicated a high level of stress related to the suspension of all clinical training activities, fear of contracting COVID-19, and dealing with the challenges of distance education.	2c/B
Cici and Yilmazel [50], Turkey, 2020	Cross-sectional study n = 322 (Women 76.4%)	BAI STAI-S		BAI: 15.2 ± 8.4 score < 16: 49.1% score 16 ≥ 50.9% STAI-S: 40.3 ± 4.9 score < 40: 48.1% score > 40: 51.9%	A significant increase in anxiety scores was found in those with a negative perspective of the profession during the pandemic (*p* < 0.05). Anxiety scores were found to be significantly higher for those with the unwillingness to practice the profession in the future (*p* < 0.05). Anxiety scores were found to be significantly higher among those with fear of being infected (*p* < 0.05).	2c/B
García González et al. [51], Spain, 2021	Cohort study n = 460 (Women 78%)	STAI		Anxiety state: score: (a) first week of confinement 26.5 ± 11.7 (b) fourth week of confinement.31.1 ± 6.9 Trait anxiety score: (a) first week: 23.9 ± 10.7 (b)fourth week 28.7 ± 6.5 Total anxiety levels: (a) first week 50.4 ± 20.8 (b) fourth week 59.9 ± 10.6	In the fourth week of confinement, STAI scores were higher than in the first week of being confined, in both males and females. Although female students had higher anxiety levels than male students in both weeks, the increase in males’ anxiety levels was higher than in females. These results are statistically significant. All students’ STAI scores increased throughout confinement, with undergraduate students in their last year of their degree program showing the highest increase in anxiety levels in the fourth week with respect to the first week of confinement (*p* < 0.001). Anxiety levels in students in confinement in a house with a garden were lower than in those living in a house without a garden throughout the confinement, with statistically significant differences observed.	2b/B
Dalcali et al. [52], Turkey, 2021	Cross-sectional study n = 283 (Women 82%)	STAI		State anxiety: 42.24 ± 10.95 Trait anxiety: 48.56 ± 9.83 51.9% of the students reported that they felt anxious in relation to the cases of COVID-19 in Turkey	High-level positive significant correlation between students’ sleep quality and state anxiety (r: 0.305, *p*: 0.000) and trait anxiety (r: 0.288; *p*: 0.000). Students’ state anxiety, trait anxiety and sleep quality did not differ by gender, age, school year, or the presence of a chronic or psychiatric disease. Students’ state anxiety differed according to their feelings associated with COVID-19, and this difference was caused by feelings of fear and anxiety (*p* = 0.039).	2c/B
Göl and Erkin [53], Turkey, 2021	Cross-sectional study n = 2630 (Women 82.1%)	GHQ-12		GHQ-12 score: 3.04 ± 2.13 71.5% of nursing students scored ≥ 2 on the GHQ-12	This result indicated that nursing students were at risk of developing mental problems. Students who slept less or ate less during the pandemic and those who thought they had symptoms of COVID-19 obtained significantly higher scores (*p* < 0.05).	2c/B
Savitsky et al. [54], Israel, 2021,	Cross-sectional study n = 244 (Women 78%)	GAD-7		(a) third week of confinement: 9.3 ± 5.6, median 9 (IQR, 5–13) (b) fifth week of confinement 7.5 ± 5.6, median 7 (IQR, 3–11.5) (*p* < 0.0001) Moderate anxiety: (a) First time point 42.8% (n = 92) (b) Second measurement 34.9% (n = 67) Severe anxiety: (a) First time point 13.0% (b) Second measurement 11.5%	In the first survey, females had significantly higher anxiety scores in comparison to males (mean 9.7 vs. 6.5); in the second survey, the mean anxiety score was similar among females and males (mean anxiety score was 7.5 among both genders). At the end of lockdown, a significant difference was found among the population groups: Muslims had the highest anxiety score compared with Jews and Christians. The mean anxiety score of Jews and Christians decreased between 2 time points, and the mean anxiety score of Muslims increased. Parents of young children, who did not feel a burden following kindergarten and school closures, had a lower mean anxiety score than those who experienced a burden. The factor of resilience was found to be significantly and negatively associated with moderate and severe anxiety. The factor of mental disengagement became more meaningful in the second survey, during the lockdown, and was found to be significantly associated with moderate anxiety at both time points and with severe anxiety during the first survey. The factor of seeking information and consultation became less meaningful in the second survey; during the lockdown and during the second survey, it became significantly associated with moderate anxiety.	2c/B
Mundakir et al. [55], Indonesia, 2020	Cross-sectional study n = 619 (Women 82.4%)	CAS		Students with anxiety n= 37 (6%) 94% nursing students did not experience anxiety due to COVID-19	Students from a private institution had a higher average of anxiety than those from a public institution (*p* = 0.034).	2c/B
Sakai et al. [56], Japan, 2022	Cross-sectional study n = 104 (Women 84.7%)	HADS		HADS score: 6.8 ± 4.2 Students scoring more than 7: 30.5% (n = 36)	Nursing students showed a significantly lower level of anxiety than students from other departments.	2c/B
Fitzgerald and Konrad [38], USA, 2021	Cross-sectional study n = 50 (Women 84%)	Anxiety Symptom Checklist		70% of students felt anxious often or very often	The effect on academics was apparent in the strong correlation between symptoms of anxiety and distress and concerns about academic matters. Respondents who described greater effects on activities and relationships during the study period most strongly questioned their ability to attain their personal goals and felt anxious or distressed about academic matters. This shows that the pandemic and the transition to all-online instruction may have had a potentially negative impact on these students’ ability to move forward in the program.	2c/B
Ozturk and Tekkas-Kerman [57], Turkey, 2022	Randomized clinical trial N = 61 (Women 100%) Case group (n = 32) Control group (n = 29)	DASS-42		Pre-test Intervention group: 11.90 ± 7.31 Control Group 9.13 ± 6.42 Post-test Intervention Group: 7.90 ± 6.70 Control Group 9.51 ± 6.58	The intervention group took eight online sessions of laughter therapy that is two sessions per week for four weeks. A laughter therapy session consists of clapping and warming-up exercises, deep breathing exercises, childlike playfulness, and laughter exercises. There was a statistically significant difference between groups in terms of depression after online laughter therapy sessions (*p* < 0.05), but there was no significant difference between anxiety, stress, and loneliness levels (*p* > 0.05).	1b/A
Al Maqbali et al. [58], Oman, UK, Saudi Arabia, Abu Dhabi, 2023	Cross-sectional study n = 918 (Women 85.7%, n = 787)	HADS		HADS-A score 10.25 ± 3.53	Students presented with stress (91.6%), anxiety (69.1%), depression (59.8%), and insomnia (73.2%). There were significant positive relationships between fear of COVID-19, stress, anxiety, depression, and insomnia.	2c/B
Bai et al. [59], China, 2021	Cross-sectional study N = 932 (Women 75.3%)	GAD-7			Among the anxiety symptoms, the Nervousness and Uncontrollable worry edge showed the strongest connection, followed by edges for Restlessness and Feeling afraid. Irritability was the most central symptom for Chinese nursing students during the COVID-19 pandemic worry.	2c/B
Cetinkaya et al. [60], Turkey, 2022	Cross-sectional study n = 291 (Women 78%)	STAI		State anxiety scale: 42.54 ± 10.68 Trait anxiety scale: 45.16 ± 9.19	Women’s trait anxiety scale scores were statistically significantly higher than those of men. The situationally anxiety scale scores of those who were not isolating themselves and those who were partially isolating were statistically significantly higher than those who were self-isolating. The state and trait anxiety scale scores of those who were happy were statistically significantly lower than the scores of those who were not happy.	2c/B
Abu Liel [61], Palestine, 2023	Cross-sectional study n = 320	DASS-21		n= 121 (37.8%)	78 (24.3%) of participants reported stress, and 72 (22.5%) reported depression. Socioeconomic status had a statistically significant effect on anxiety, stress, and depression. In addition, place of residence had a statistically significant effect on anxiety.	2c/B
Masha’al et al. [62], Jordan, 2022	Cross-sectional study n = 282 (Women 74.1%)	GAD-7		No anxiety (n = 83) 29.4% Mild anxiety (n = 100) 35.5% Moderate anxiety (n = 60) 21.3% Severe Anxiety (n = 39) 13.8%	Nursing students experienced significant COVID-19 infection-related anxiety upon returning to on-campus learning. Dysfunctional coping strategies were associated with increased levels of anxiety. Female students and students who did not take personal protective measures to prevent infection with the virus reported higher anxiety than those who were not afraid of infection.	2c/B
Dubovi et al. [63], Israel, 2022	Cross-sectional study n = 135 (Women 88%)	STAI		42 ± 13.0 56% (n = 76) nursing students scored > 40 16% (n = 22) of nursing students scored > 55 cut-off denoting severe anxiety	58% of all the research participants experienced clinically significant anxiety levels; the anxiety prevalence among nursing students was similar to that of the general population. The prevalence of anxiety was significantly higher among nurses compared to nursing students and the general public.	2c/B
Akpinar et al. [64], Turkey, 2022	Cross-sectional study n = 1037 (Women 61.2%)	CAS		2.49 ± 3.85	There was a positive and moderate level of correlation between the scores on the cyberchondria scale and internet addiction scale and the COVID-19 anxiety scale. There was a linear and positive relationship between internet addiction and COVID-19 anxiety. Also, there was a relationship between cyberchondria levels and COVID-19 anxiety. Students with high COVID-19 anxiety have higher levels of cyberchondria. People were exposed to excessive information about COVID-19 during the pandemic, and this worsened anxiety, which then could lead to cyberchondria. In the association between internet addiction and cyberchondria severity, COVID-19 anxiety acted as a mediator. Measures such as isolation and social distancing to prevent the spread of the virus during the COVID-19 pandemic led people to spend more time on the internet. In addition to the above, the cyberchondria levels of students who reported that they used the internet as a source of reference for health information when they had a medical condition were significantly higher in our study.	2c/B
Meneses and Andrade [65] Brazil, 2024	Cross-sectional study n = 206 (Women 85.9%)	DASS-21		Normal n = 73 Mild n = 9 Moderate n = 30 Severe/very severe N = 94	The prevalence of moderate to extremely severe symptoms of depression, anxiety and stress among nursing students with smartphone dependence was 64.6%, 64.5% and 63.1%, respectively.	2c/B
Masha’al, Rababa et al. [66], Jordan, 2024	Cross-sectional study n = 282 (Women 74.1%)	GAD-7		8.06 ± 5.49 indicating mild level of anxiety	Nursing students who had high levels of emotional intelligence also showed lower anxiety level.	2c/B
Zukhra et al. [67], Indonesia, 2021	Cross-sectional study n = 224 (Women 90.7%)	ZSAS		Mild to severe anxiety score: 35.3% (n = 87)	Increased anxiety levels were correlated to outdoor activities, being a female student, living in a red zone or being younger than 20 years.	2c/B

## Data Availability

Not applicable.

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
