# Peer review of "Anxiety in Nursing Students during the COVID-19 Pandemic: Systematic Review and Meta-Analysis"

_healthcare, 2024, doi:10.3390/healthcare12161575_

Round 1

Reviewer 1 Report

Comments and Suggestions for Authors

The manuscript utilized a meta-analysis method to investigate the influence of the Covid-19 pandemic on the development of anxiety in nursing students and also detect the risk factors for developing anxiety. Here I'd like to give several comments and suggestions that may be considered by the author(s) to enhance the rationale and improve the flow of writing. First, in the Introduction section, more detailed writing should be added to build explicit linkage between Covid-19 pandemic and anxiety in nursing students to enhance the rationale of the writing of the manuscript and also give prominence to the significance of the research, i.e., research gap that the author(s) wanted to fill, as a large number of studies have probed into Covid-19 and its impact on the mental health in general, and anxiety in particular, in students. Besides, a theoretical framework should be put forward to guide the analysis of the data and hence the discussion of the results. In addition, the author(s) may consider include research published in the year of 2023, as quite a number of empirical research into nursing students' mental health was put forward in this year. What's more, the author(s) may also pay attention to the bias of the data collected. For instance, the author(s) may compare the Covid-19 impacts as revealed by the published research with those documented in non-published articles. Finally, the practical and research implications of the research should be well organized and added to the end of the manuscript. The author(s) are also suggested to list writings that help inspire future research. 

Comments on the Quality of English Language

Professional editing and proofing are required to enhance the language use and the writing of the manuscript.

Author Response

Dear reviewer

Thank you for reviewing the manuscript and for your recommendations and comments. The response to each comment and the changes in the manuscript have been highlighted in yellow.

Kind regards

Reviewer 1

The manuscript utilized a meta-analysis method to investigate the influence of the Covid-19 pandemic on the development of anxiety in nursing students and also detect the risk factors for developing anxiety. Here I'd like to give several comments and suggestions that may be considered by the author(s) to enhance the rationale and improve the flow of writing. ‘

1-First, in the Introduction section, more detailed writing should be added to build explicit linkage between Covid-19 pandemic and anxiety in nursing students to enhance the rationale of the writing of the manuscript and also give prominence to the significance of the research, i.e., research gap that the author(s) wanted to fill, as a large number of studies have probed into Covid-19 and its impact on the mental health in general, and anxiety in particular, in students.

We have reviewed it.

-Besides, a theoretical framework should be put forward to guide the analysis of the data and hence the discussion of the results.

We have check it.

-In addition, the author(s) may consider include research published in the year of 2023, as quite a number of empirical research into nursing students' mental health was put forward in this year.

We have added more articles.

-What's more, the author(s) may also pay attention to the bias of the data collected. For instance, the author(s) may compare the Covid-19 impacts as revealed by the published research with those documented in non-published articles.

We had tried it, but it was impossible at the end

-Finally, the practical and research implications of the research should be well organized and added to the end of the manuscript. The author(s) are also suggested to list writings that help inspire future research. 

We have added it in the text.

Reviewer 2 Report

Comments and Suggestions for Authors

The study represents an important study to make review and meta-analysis on the influence of the COVID-19 pandemic on anxiety in nursing students and the related factors. Generally, the manuscript is well-written with a clear structure. The introduction provides a clear context for study. The methods section is well-described. The results are clearly summarized, highlighting higher anxiety scores during the pandemic and identifying specific risk and protective factors. The conclusions effectively summarize the findings and their implications for nursing students' mental health. I’d appreciate that the authors may consider some points.

Abstract: "around 50%" (line 22) sounds ambiguous.

Line 18-20: due to space limitation, the search equation could be separated from the abstract, as it is available in the main text.

Research questions could be added, such as: What is the prevalence of anxiety among nursing students during the COVID-19 pandemic? What are the primary factors contributing to increased anxiety among nursing students during the pandemic? What protective factors mitigate anxiety?

Method: any models or software used? heterogeneity assessments included? Was there any model/method used for to calculate an overall pooled prevalence estimate?

Figure 2 (line 309): spelling: Forest plot.

Discussion: the authors may consider discussing different anxiety scales described in the findings and possibilities of bias.

In the introduction, the authors mentioned stress and depression. The linkage between stress, anxiety and depression within the context of this study may need to be discussed.

Spiritual factor (line 338) is very intrigue and may need to be discussed.

Conclusion: it could be more concise and make into one paragraph.

Author Response

Dear reviewer

Thank you for reviewing the manuscript and for your recommendations and comments. The response to each comment and the changes in the manuscript have been highlighted in yellow.

Kind regards

Reviewer 2

The study represents an important study to make review and meta-analysis on the influence of the COVID-19 pandemic on anxiety in nursing students and the related factors. Generally, the manuscript is well-written with a clear structure. The introduction provides a clear context for study. The methods section is well-described. The results are clearly summarized, highlighting higher anxiety scores during the pandemic and identifying specific risk and protective factors. The conclusions effectively summarize the findings and their implications for nursing students' mental health. I’d appreciate that the authors may consider some points.

-Abstract: "around 50%" (line 22) sounds ambiguous.

We have specified it in the text.

-Line 18-20: due to space limitation, the search equation could be separated from the abstract, as it is available in the main text.

We agree with your comment. We have deleted the search equation.

-Research questions could be added, such as: What is the prevalence of anxiety among nursing students during the COVID-19 pandemic? What are the primary factors contributing to increased anxiety among nursing students during the pandemic? What protective factors mitigate anxiety?

We have added it in the text.

-Method: any models or software used? heterogeneity assessments included? Was there any model/method used for to calculate an overall pooled prevalence estimate?

The statistical software Stats Direct was used to perform the meta-analysis.

We have added all in the text.

-Figure 2 (line 309): spelling: Forest plot.

Thanks a lot for this comment. We have made that spelling change.

Discussion: the authors may consider discussing different anxiety scales described in the findings and possibilities of bias.

We had decided not to add this consideretions to this manuscript cause it was quite long.

In the introduction, the authors mentioned stress and depression. The linkage between stress, anxiety and depression within the context of this study may need to be discussed.

Spiritual factor (line 338) is very intrigue and may need to be discussed.

We have discussed both questions.

Conclusion: it could be more concise and make into one paragraph.

We have reviewed the conclusion.

Round 2

Reviewer 1 Report

Comments and Suggestions for Authors

The authors have addressed the concerns regarding the manuscript.

Author Response

Thanks for the comments, we have already reviewed the manuscript